# General Anesthetics in Cancer Surgery: Can Anesthesiologists Help the Patient with More than a Safe Sleep

**DOI:** 10.3390/medicina58091156

**Published:** 2022-08-25

**Authors:** John Michael Bonvini

**Affiliations:** 1Department of Anesthesiology, Ars Medica Clinic, Via Grumo, 16, 6929 Gravesano, Switzerland; johnmichael.bonvini@uzh.ch; 2University of Zurich, Rämistrasse, 71, 8006 Zurich, Switzerland

**Keywords:** cancer surgery, general anesthetics, propofol, volatile anesthetics

## Abstract

Most patients suffering from neoplastic diseases will at some point during their illness be approached surgically. Surgery itself may be unfortunately responsible for tumor proliferation and metastatic spread. With the perioperative period increasingly becoming a focus of research in anesthesia, anesthesiologists have looked at the chance to influence cancer progression based on their choice of anesthesia regimen and strategy. Many anesthetic agents have been investigated for their potential impact on the course of cancer disease. There is an abundance of retrospective studies and very few prospective ones that tackled this issue. The aim of this article is to review the current state of the evidence on general anesthesia involving volatile and intravenous agents as substrates, focusing on halogenated inhalational agents and propofol, to guide clinical decision making in assessments of the best practice for perioperative management of cancer surgery.

## 1. Introduction

Neoplastic disease is a one of the major causes of mortality. Worldwide, over 19 million new cancer cases and 10 million cancer-related deaths were estimated for 2020, and for the first-time, female breast cancer, with 2.3 million new cases (11.7% of all new cases), surpassed lung cancer (11.4%), followed by colorectal disease (10%) and prostate cancer (7.3%) [1].

Most solid tumors are at some stage approached surgically, and over 80% of new cancer cases will require surgery at least once if not several times during the course of the disease [2].

General anesthetics (GA) are used to facilitate surgical procedures and can be generally divided in volatile (inhalational) and intravenous agents. Both induce, as a primary effect, unconsciousness in the patient. The exact mechanism of actions, even after over decades of safe clinical use, are still being investigated [3].

Perioperative medicine has become a major focus for modern anesthesiologists [4] and has sparked interest as a possible anesthesiologic strategy in the surgical management of cancer patients since it is well known that, all over the world, drugs used in operating theaters to induce unconsciousness on a daily basis have more effects than they are primarily used for.

Among these many effects, of particular interest are the effects on extracellular matrix (ECM)-degrading proteins such as matrix metalloproteinases (MMP). These proteins are potent remodelers of the ECM, and their abnormal expression bears the potential of enhancing metastatic disease [5]. The effects of GA on microRNAs (MiR) are also being increasingly investigated. MiR are small non-coding RNAs that modulate gene expression and might be heavily dysregulated in cancerous disease [6]

## 2. Volatile Anesthetics

Volatile anesthetics have shown a beneficial effect in ischemia reperfusion injury [7,8,9]. Through the suppression of NK cells, macrophages, and neutrophils, they modulate the innate immune response as well as the adaptive immune system by increasing lymphocyte apoptosis while inhibiting their proliferation [10,11,12]. Unfortunately, these same effects can, however, be detrimental in other settings, such as in general anesthesia for cancer surgery [10].

The results of in vitro studies using volatile anesthetics in cancer cells are heterogeneous. The in vitro exposure of renal carcinoma cells to sevoflurane increased cancer resistance to the chemotherapeutic agent cisplatin, and increased cell migration and viability [13]. Interestingly, in the same in vitro setting but using non-small cell lung carcinoma (NSCLC) cells, sevoflurane reduced the cell viability and increased the sensitivity to cisplatin without increasing cell migration [13].

Exposure of human ovarian carcinoma cells to halogenated inhalational anesthetics (isoflurane, sevoflurane, and desflurane) enhanced the expression of numerous genes associated with metastatic proliferation [14] and, with isoflurane, led to an increase in markers associated with malignant capabilities such as cell proliferation, migration, as well as angiogenesis [15]. In contrast, volatile agents have been shown to *reduce* ECM breakdown through the down-regulation of Matrix metalloproteinase 9 (MMP-9), thus reducing the metastatic cancer cell potential [16].

## 3. Propofol

The intravenous general anesthetic propofol has also been investigated regarding its potential effects on cancer cells [17].

In human pancreatic cancer cells, propofol has been found to decrease cell proliferation and migration as well to increase apoptosis rates, possibly through the upregulation of microRNA miR-133a [18]. Propofol also potentiated growth inhibition and apoptosis induced by the chemotherapeutic agent gemcitabine and downregulated NF-kB DNA binding activity, hence counteracting its anti-apoptotic activity [19].

In ovarian cancer cells, it modulated microRNA miR-let-7i, inhibited cell proliferation, and increased apoptosis [20]. Its activation of miR-9 was associated with an inhibition of metalloproteinase MMP-9 involved in the degradation of the extracellular matrix; was associated with the invasive potential of cancer cells; and modulated the NF-kB pathway, thus enhancing apoptosis [21]. Some ovarian cell lines show different resistances to chemotherapeutic agents, one of which is paclitaxel. Interestingly, propofol inhibited cell invasion and proliferation in paclitaxel-resistant but not in paclitaxel-sensitive cell lines, while apoptosis was enhanced in both sensitive and resistant cell lines [22].

Looking at hepatocellular carcinoma (HCC), propofol also efficiently inhibited the growth of HCC cells and activated both caspase-8 and caspase-9, suggesting that both intrinsic and extrinsic pathways induce apoptosis. This is thought to occur through the modulation of miR-199a microRNA [23]. Propofol also reduced the adhesion ability of HCC cells and downregulated the expression and activity of MMP-9 associated with the facilitation of HCC cell invasion in neighboring tissues [24].

In lung adenocarcinoma cells, the inhibition of MMPs via propofol also resulted in diminished cell invasion and migration [25]. Similarly, in non-small cell lung cancer, propofol showed a reduction in the mRNA expression of various MMPs as well as the inhibition of cell adhesion motility and migration [26]. Propofol in NSCLC cells effectively suppressed the effects of inflammation, which is associated in the tumor microenvironment, and promotes cancer aggressiveness, together with a significant reduction in Hypoxia inducible factor 1 alpha, the high expression of which is associated with poor prognosis [26]. Furthermore, propofol was also found to suppress NSCLC cell growth, to accelerate apoptosis through the miR-21/PTEN/AKT pathway in vitro, and to effectively inhibited tumor growth in vivo in an animal model [27].

On the other hand, recent work has shown that propofol promoted tumor metastasis, probably through its affinity for the GABAAR receptor, downregulating the expression of the tripartite motif protein family TRIM21 and subsequently increasing the expression of the Src protein [28]. It is in fact known that the expression of Src is associated with the metastatic potential of cancer cells since it promotes the epithelial–mesenchymal transition that occurs during tumor extravasation [29] and its inhibition suppresses the adhesion of lung cancer cells [30].

In breast cancer, propofol has been shown to significantly downregulate microRNA miR-24, the overexpression of which promotes cell proliferation and inhibits apoptosis [31]. On the other hand, the GABA agonism of propofol has been shown to increase the migration of breast carcinoma cells by reorganizing the actin cytoskeleton [32] and its antioxidant properties have been associated with increasing cell proliferation, through the inhibition of p53, a recognized tumor-suppressor gene and up-regulator of cell apoptosis [33].

## 4. Clinical Studies

There is an abundance of retrospective clinical trials assessing potential differences in general anesthesia regimes in cancer surgery, with many of them favoring the use of propofol, e.g., as in colon cancer [34], esophageal cancer [35], hepatectomy for HCC [36], radical prostatectomy [37], and breast cancer [38]. In other retrospective studies where no difference in overall mortality was found, a lower recurrence rate was associated with the use of propofol [39,40], while no difference either in overall or recurrence-free survival was found in NLCLC [41] and in breast cancer [42,43].

Since immunomodulation plays a significant role in immunosuppression, inflammation, and cancer recurrence [44], a recent metanalysis looked at interleukin (IL) 6, IL-10, tumor necrosis factor, and C-reactive protein as surrogate markers for perioperative inflammation. The authors found no differences between the two general anesthetic techniques, suggesting no definitive impact of the choice of general anesthetic on the inflammatory response and its associated cancer aggressiveness [26,45].

On the other hand, in a large retrospective study over a three-year period, looking at long-term survival in patients, Wigmore et al. found an increased hazard ratio (1.46, 95% CI 1.29 to 1.66) for patients undergoing surgery under general inhalational anesthesia [46]. Similarly, Yap et al. suggested in their systematic review that propofol-based general anesthesia may be associated with better recurrence free and overall survival when compared with volatile anesthetics [47], acknowledging, however, that the inherent limitations of the 10 studies included in their review commanded larger, well-designed prospective studies to corroborate these findings.

In fact, very few prospective clinical studies have tackled the question of whether propofol clinically offers benefits over inhalational anesthesia in cancer surgery.

Oh et al. hypothesized that propofol would be less immunosuppressive than sevoflurane in breast surgery; looking at regulatory T cells, Oh et al. found that there was no difference in the tumor-cell-killing capabilities of natural killer- and cytotoxic T-cells [48].

Yan et al. showed in patients undergoing breast cancer surgery that vascular endothelial growth factor (VEGF) serum concentrations where higher after general anesthesia with sevoflurane when compared with propofol but, despite the increase in this angiogenesis promoting marker, found no difference in the recurrence-free survival when comparing the two anesthesiologic strategies [49].

In metastatic disease, one factor affecting the distant spread of the disease is the number and characteristics of circulating tumor cells (CTC) [50,51,52,53]. In a recent work, the influence of the choice of general anesthetic on the number of CTCs in vivo was assessed in women undergoing surgery for breast cancer as an independent prognostic factor. In a parallel controlled trial, et al. randomized 210 women undergoing primary breast cancer surgery to receive either sevoflurane or propofol based general anesthesia. Hovaguimian et al. found no difference in the median number of circulating tumor cells assessed at 0, 48, and 72 h. Interestingly, they noted that administrating sevoflurane led to a significant increase in the maximal tumor cell counts postoperatively, but no association was found between circulating tumor cell counts and natural killer cell activity [54].

## 5. Conclusions

The question of which anesthesia regimen is better suited for patients undergoing cancer surgery cannot be definitively answered without further prospective clinical research. Most of the evidence at this time seems to favor propofol, but most probably, this will not become a “magic cancer bullet”. A thorough evaluation of the risk–benefit profiles of inhalational/intravenous strategies for individual patients is key in choosing a tailored anesthesiologic regimen.

After all, informed, individual, and precise perioperative management is the best help an anesthesiologist can offer its patients for a safe sleep.

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
