# Peer review of "General Anesthetics in Cancer Surgery: Can Anesthesiologists Help the Patient with More than a Safe Sleep"

_medicina, 2022, doi:10.3390/medicina58091156_

Round 1

Reviewer 1 Report

In his manuscript, entitled “General anesthetics in cancer surgery: can anesthesiologists 2 help the patient with more than a safe sleep” J. M. Bonvini is providing a brief narrative review of the literature regarding the role of volatile and IV anesthetics on outcome in patients undergoing cancer surgery.

The overall idea of the manuscript is interesting, the manuscript is well-written and adheres to all journal standards. There are only very few issues, which I would kindly ask the author to address before this manuscript can be published:

Overall: Please be a little more elaborate regarding the introduction of the main targets of the respective anesthetic agents. Key players like MMP-9 or the different miRNAs should be explained at least with one sentence in order to increase the readers’ ability to understand the importance of the mentioned effects.

Volatile anesthetics: Since the effects of propofol on MMP-9 expression are mentioned, maybe you should consider also mentioning the effects of volatile anesthetics on this important enzyme (e.g. PMID: 24989774).

Author Response

We thank the reviewer for the precious inputs.

Overall: Please be a little more elaborate regarding the introduction of the main targets of the respective anesthetic agents. Key players like MMP-9 or the different miRNAs should be explained at least with one sentence in order to increase the readers’ ability to understand the importance of the mentioned effects.

To enhance users’ readability, we followed the valuable hints of the reviewer and added in the introduction a general reference about the mechanistic common effects of general anesthetics and a small general reference to MMPs and MiR.

 Volatile anesthetics: Since the effects of propofol on MMP-9 expression are mentioned, maybe you should consider also mentioning the effects of volatile anesthetics on this important enzyme (e.g. PMID: 24989774).

Also, the effects of volatile on MMPs were added in the Volatile section

Reviewer 2 Report

Well written review paper.

Cancer is a leading health problem worldwide. Data from retrospective studies suggests that anesthesia and anesthetics may contribute significantly to pro-tumor environment and affect long-term outcomes after cancer surgery. The lack of prospective research on this topic is evident.

The author performed a detailed analysis of the research results examining the influence of anesthesia techniques and anesthetic agents (volatile anesthetics and propofol)  on the course of cancer after various types of cancer surgery

It addresses important issues with both practical and theoretical implications and it provides good guidance for prospective research on this topic.

The author has done a good job.

Author Response

We thank the reviewer for the positive and stimulating comments.